# Is Monetary Policy a Driver of Cryptocurrencies? Evidence from a Structural Break GARCH-MIDAS Approach

**Md Samsul Alam [1], Alessandra Amendola [2], Vincenzo Candila [2] and Shahram Dehghan Jabarabadi [2,*]**

[1] College of Business, Law and Social Sciences, University of Derby, Derby DE22 1GB, UK; m.alam@derby.ac.uk

[2] Department of Economics and Statistics, University of Salerno, 84084 Fisciano, Italy; alamendola@unisa.it (A.A.); vcandila@unisa.it (V.C.)

\* Correspondence: shahram.dehghan.j@icloud.com

**Abstract:** The introduction of Bitcoin as a distributed peer-to-peer digital cash in 2008 and its first recorded real transaction in 2010 served the function of a medium of exchange, transforming the financial landscape by offering a decentralized, peer-to-peer alternative to conventional monetary systems. This study investigates the intricate relationship between cryptocurrencies and monetary policy, with a particular focus on their long-term volatility dynamics. We enhance the GARCH-MIDAS (Mixed Data Sampling) through the adoption of the SB-GARCH-MIDAS (Structural Break Mixed Data Sampling) to analyze the daily returns of three prominent cryptocurrencies (Bitcoin, Binance Coin, and XRP) alongside monthly monetary policy data from the USA and South Africa with respect to potential presence of a structural break in the monetary policy, which provided us with two GARCH-MIDAS models. As of 30 June 2022, the most recent data observation for all samples are noted, although it is essential to acknowledge that the data sample time range varies due to differences in cryptocurrency data accessibility. Our research incorporates model confidence set (MCS) procedures and assesses model performance using various metrics, including AIC, BIC, MSE, and QLIKE, supplemented by comprehensive residual diagnostics. Notably, our analysis reveals that the SB-GARCH-MIDAS model outperforms others in forecasting cryptocurrency volatility. Furthermore, we uncover that, in contrast to their younger counterparts, the long-term volatility of older cryptocurrencies is sensitive to structural breaks in exogenous variables. Our study sheds light on the diversification within the cryptocurrency space, shaped by technological characteristics and temporal considerations, and provides practical insights, emphasizing the importance of incorporating monetary policy in assessing cryptocurrency volatility. The implications of our study extend to portfolio management with dynamic consideration, offering valuable insights for investors and decision-makers, which underscores the significance of considering both cryptocurrency types and the economic context of host countries.

**Keywords:** structural break; GARCH-MIDAS; cryptocurrency; monetary policy

**JEL Classification:** E52; C53; C52; G11

## 1. Introduction

The application of blockchain technology as an advanced database mechanism with a decentralized, tamper-proof system for recording transactions has subsequently created a high number of cryptocurrencies. The implication of cryptocurrency in the global financial markets and its growing share in institutional investor participation, mainly throughout 2020–2021, has encouraged investors and hedge funds to be interested in cryptocurrencies (Fletcher 2021).

Transitioning from the global financial market perspective, it is essential to explore the multifaceted role that cryptocurrencies play. The creation of cryptocurrency led to the

impressive expansion of liquidity in the path of macroeconomic stability (Nekhili et al. 2023). Cryptocurrencies retain some of the gold hedging capabilities against market risk and, therefore, they can be considered an instrument for risk-minimizing investments (Dyhrberg 2016). In addition, fewer control mechanisms over cryptocurrencies, such as a central bank and interest rate, have led the blockchain technology to reduce the transaction cost of the banking system, especially in terms of cross-border transactions (Hassani et al. 2018). Cryptocurrencies also equipoise the excess demand for money through market mechanisms when they are adopted, which can be an effective tool if there is no limitation for expanding the money stock as a safe unit of currency among eager individuals (Caton 2020). These aspects underline the potential benefits of cryptocurrencies as instruments of economic stability and efficiency.

However, alongside their potential benefits, cryptocurrencies also pose risks and challenges. They can act as: a risk and threat to global financial stability (FSB 2022); the risk source for traditional financial markets and a net to transmit the dynamic spillovers (Li et al. 2023); the cause of losing investor confidence and financial stability in emerging markets due to the lack of policies and regulations (FSB 2022); and a poor hedging instrument in portfolios (Charfeddine et al. 2020). Moreover, cryptocurrencies, mainly Bitcoin, are speculative investments that have an inverse correlation with the stock market and pro-cyclical behavior (Baur et al. 2018; Conrad et al. 2018).

Despite the disagreement regarding the impact of cryptocurrencies on economic variables, which occurs because of different views and regulations in various countries and institutions facing cryptocurrencies (Jiménez-Serranía et al. 2022), stability solutions and mechanisms outside of the economic system, such as monetary policy, encourage the development of blockchain technology and cryptocurrencies because they may supply liquidity for demand alteration for money (Caton 2020). However, the way that cryptocurrencies behave with other economic variables and economic stability depends on the economic and political structure of the studied population. According to Štefan Lyócsa et al. (2020), Bitcoin volatility was affected by macroeconomic news related to Bitcoin regulation. They applied a statistical model (quantile regression) to the measures of Bitcoin variability and jump components to examine how Bitcoin reacted to macroeconomic events and news (mostly from the USA). Furthermore, events and geopolitical crises affect the cryptocurrency market; however, the causes of the fluctuations are not clear (Alexakis et al. 2024)[1]. Therefore, it is crucial to consider regional and structural differences when assessing the relationship between cryptocurrencies and monetary policy.

Whatever the nature of cryptocurrency, it is of great interest to investors to have accurate volatility estimates and forecasts. Cryptocurrencies and their volatility contain a high volume of uncertainty, which influences the effectiveness of predictions and decision-making in this market. A potential driver of the cryptocurrency's volatility is monetary policy (Ma et al. 2022). The way cryptocurrencies are expressed has piqued the interest of monetary authorities, in that it suggested the possibility of issuing the cryptocurrencies directly by central banks (Lucarelli and Gobbi 2023).

It is worth mentioning that the relationship between monetary policies and crypto markets is bidirectional; however, our study addresses the impact of monetary policy on cryptocurrencies to avoid the complexity of evaluations. Regional differences, various blockchain mechanisms, and cryptocurrencies may have distinct impacts on monetary policy and welfare. For instance, the comparison between Proof-of-Work (PoW) and Proof-of-Burn (PoB) highlights differences in reducing volatility and increasing welfare (Saleh 2018). Cryptocurrencies also boost worries for governments about the prevailing effect of the new cryptocurrencies on the overall stability of monetary and financial systems (Charfeddine et al. 2020). Generally, incorporating the Economic Policy Uncertainty Index into Bitcoin volatility models enhances their predictive performance using high-frequency data (Yu 2019). Corbet et al. (2017) studied how Bitcoin responds to monetary policy and found that Bitcoin behaves like traditional fiat currencies and is affected by them. They used a statistical model (GARCH (1,1)) for Bitcoin and a measure of how the domestic

currency changes against the trade-weighted index of the domestic currency in a basket of USD, EUR, CNY, and GBP. These nuances underscore the need for a comprehensive analysis that considers different cryptocurrencies and their underlying technologies.

Regulations and stability mechanisms can influence the efficiency of cryptocurrencies and blockchain technology toward economic stabilization (Caton 2020), which is one of the reasons for disagreement concerning the impact of monetary policy on cryptocurrencies. For instance, Bundesbank (2021) reports the impact of the European Union's monetary policy system on crypto tokens. The research utilized the average return volatility in the time windows around the announcements of monetary policy decisions by the European Central Bank (ECB) Governing Council. Based on a vector autoregression (VAR), Deutsche Bundesbank figured out that the impulse of the European Union's monetary policy system significantly influences cryptocurrencies in comparison to shares and exchange rates. In a more recent study, Corbet et al. (2020) covered the volatility spillover effect of the USA's monetary policy announcement on cryptocurrencies. The consideration of 58 digital assets, the USD nominal broad dollar index, and eight interest rate changes and quantitative easing (QE) announcements made by the United States Federal Market Open Committee show how the features of cryptocurrencies impact relationships.

On the other hand, Vidal-Tomás and Ibañez (2018) do not detect a significant signal from monetary policy events of the Federal Reserve System, European Central Bank, Bank of Japan, and Bank of England to Bitcoin's returns. Demir et al. (2018), according to an investigation on economic policy uncertainty and Bitcoin returns, found that the Bitcoin returns will be negatively impacted by economic policy uncertainty. However, there may not be an underlying reason indicated from the output. Somehow, price uncertainty and involved persons have more influence on the volatility of the market in comparison to policy uncertainty (Lucey et al. 2022). In addition, cryptocurrency should have zero inflation, which prevents the government from debasing fiat money. This means that models where the money supply is endogenous and costly can allow cryptocurrencies and fiat money, which are influenced by monetary policy, to coexist (Yu 2023).

Despite the rich literature on the topic, other authors have mainly concentrated on Bitcoin and paid less attention to other cryptocurrencies with different background technologies and to various behaviors in front of the economic variables, especially macroeconomic decisions. Currency-based digital assets are likely more susceptible to the USA's monetary policy announcements, while applications or protocol-based digital assets are immune to these shocks. Similar differences are found for mineable and non-mineable currencies, meaning that the response to various types of uncertainty for some digital assets would be distinct from that of Bitcoin. In addition, previous authors fail to investigate other countries' economies, and they interpret them according to the United States Federal Reserve System. Moreover, they use Bitcoin as a proxy for USD because Bitcoin has a high share of the cryptocurrency market.

Koutmos (2018) used a wide range of cryptocurrencies, which reveals the time-varying nature of spillovers among cryptocurrencies. He found that employing other cryptocurrencies besides Bitcoin ensures a robust relationship between monetary policies and cryptocurrencies. Nguyen et al. (2019) compare the generalized impact of policy rates on cryptocurrency prices in the monetary tightening versus easing regimes for the USA and China. They detect an increase in the cryptocurrency price in the tightening Chinese monetary policy regimes; however, there is no relation between interest rate and cryptocurrencies for both countries. The research utilizes the United States Federal Reserve System's Open Market Operations (OMO) announcement and target rate for the monetary policy of China and the USA. Moreover, there is no evidence of volatility spillover in the variables.

Helmi et al. (2023) employ a non-linear, time-varying, vector-autoregressive framework to investigate the impact of the Central Bank Digital Currency Uncertainty Index (CBDCU) on financial and cryptocurrency markets. They detect most of the variance in cryptocurrency uncertainty from the CBDCU shocks. In a recent study, Yen et al. (2023) studied the correlation of Bitcoin with other cryptocurrencies in the presence of economic

policy uncertainty (EPU) and how it changes over time concerning the behavior of various economics with the cryptocurrency (friendly, unfriendly). They found that the Pearson correlation evaluations in the presence of an increase in EPU empower dependency at the global level and in crypto-friendly countries. Marmora (2022) conducted an event study using panel data from 26 emerging countries and local Bitcoin information. His study showed that the demand and trade volume of Bitcoin increased after the monetary policy announcement for governments that did not respond to inflation threats.

In summary, the literature on the impact of monetary policies on cryptocurrency fluctuations is still inconclusive, especially with respect to long-term fluctuations. The existing literature primarily focused on Bitcoin and United States policies. Furthermore, Karau (2023) detected that the impact of monetary policy needs time to take effect on Bitcoin returns. He utilized high-frequency Bitcoin data in an event study and a structural VAR analysis in the presence of the monetary policy of the USA (Federal Open Market Committee). To further investigate the relationship between variables in the underlying structure of a time series, it is crucial to consider both the long-run impact and important events related to monetary policy. Additionally, we would like to investigate sudden shifts in this relationship caused by events, known as structural breaks.

To fill this gap, we contribute to the literature by enhancing a Generalized AutoRegressive Conditional Heteroskedasticity Mixed Data Sampling (GARCH-MIDAS, (Engle et al. 2013)) framework with respect to structural breaks in order to investigate whether monetary policy influences cryptocurrency's volatility. Compared to previous studies like Marmora (2022) and Vidal-Tomás and Ibañez (2018), which utilized event study methodology or low-frequency data for both monetary policy variables and cryptocurrencies and selected sampling rates based on the availability of monetary policy data, the GARCH-MIDAS model partitions the overall volatility into two components: a long-run term, which varies according to the additional, low-frequency volatility determinant, and a short-run part, which instead varies daily. Therefore, the main appeal of the GARCH-MIDAS model is to solve the frequency issue, that is, to jointly use variables observed at lower and higher frequencies. The use of GARCH-MIDAS is not new in the cryptocurrency literature. For instance, Conrad et al. (2018) investigate Bitcoin volatility as a driver of global economic activity and the United States stock market. In the multivariate context, Candila (2021) shows that monthly Google Trends are useful variables for calculating the conditional covariance matrix of a panel of cryptocurrencies. Beside the cryptocurrency topics, the GARCH-MIDAS model has been largely used to determine the impact of macro-economic variables on the long-run volatility of interest (Amendola et al. 2017; Conrad and Kleen 2020; Pan et al. 2017; Su et al. 2017).

Generally, monetary policies are the tools governments use for economic stabilization. However, unexpected national and international issues affect the mechanisms of monetary policy. These unpredicted events raise the complexity of estimation and detection between data unit roots and structural breaks (Allaro et al. 2018; Perron 2005). Owyang and Wall (2004) detected various mechanisms in the shock propagation of monetary policy due to structural breaks with a regional vector autoregression (VAR) model. Likewise, a number of studies (for instance, Francis et al. 2020; Inoue and Okimoto 2008; Lee and Son 2013, among others) confirmed the importance of structural breaks in investigating monetary policy and its shocks. Different from the work of Pan et al. (2017), we assume that the low-frequency variable, that is, monetary policy, may have a structural break. If detected, the structural break (SB) date could be used to estimate two GARCH-MIDAS models, one before and one after the break. Therefore, the resulting SB-GARCH-MIDAS model is useful in cases where the low-frequency variables exhibit a structural break. The models obtain the SB through the Quandt likelihood ratio (QLR) test (Quandt 1960), which is a modification of the Chow test (Chow 1960) and a common practice when the date of the break is not known (Éric Racicot and Théoret 2016; Seok et al. 2023; Yang et al. 2023). The monetary policy is assumed to follow an autoregressive process, as also done by Vidal-Tomás and Ibañez (2018). Even though the assumption that only one break may exist could be restrictive,

the single-break evaluation presents theoretical consistency and efficiency even within multi-break cases (Bai and Perron 1998, among others). Moreover, multiple break dates cause short intervals, especially in the case of low-frequency macroeconomic data, which affect the Chow test output.

In the empirical application, we estimate the volatility of three cryptocurrencies (Bitcoin, Binance Coin, and XRP). Bitcoin, the pioneering cryptocurrency (which emerged in 2009), is a decentralized digital currency built on blockchain technology. Recognized for its limited supply of 21 million coins, Bitcoin works independently of central authorities, presenting a secure and pseudonymous trade. Binance Coin (BNB) (which emerged in 2017) is an Ethereum-based token that migrated to Binance's native blockchain, Binance Chain. Moreover, it serves diverse functions within the Binance ecosystem, such as fee discounting and participation in token sales. Finally, XRP is a digital currency associated with Ripple, which emerged in 2012 to facilitate fast and cost-effective cross-border transactions. Importantly, XRP's connection with governments and authorities has been subject to scrutiny, with ongoing discussions about its classification and regulatory framework.

Furthermore, the study employed M3 data for both the USA and South Africa, with the latter country implementing phased and structured regulation for handling crypto assets due to concerns related to money laundering, terrorism financing, and tax evasion. The SB-GARCH-MIDAS model is evaluated against some popular alternatives. Using the monetary policies of two economies (the USA and South Africa), some interesting findings are: (i) the monetary policies of the USA and South Africa impact the long-run volatility of the cryptocurrencies under investigation; (ii) based on the structural breaks in the monetary policy, the SB-GARCH-MIDAS model generally outperforms the competing models used here. By considering a wide range of cryptocurrencies and potential structural breaks in monetary policy, this research aims to provide a more comprehensive understanding of the complex relationship between monetary policies and cryptocurrencies.

The rest of the paper is structured as follows: Section 2 presents the utilized methodology and illustrates the proposed model. Section 3 provides the empirical analysis. Finally, Section 4 expresses the concluding remarks.

## 2. Materials and Methods

Let $r_{i,t}$ be the log-returns of the asset under investigation on day $i$ (high-frequency period) of period $t$ (low-frequency period). The GARCH-MIDAS model assumes that the whole volatility is divided into two components: a short-run and a long-run part. In this framework, the conditional hereroskedastic model for $r_{i,t}$ is:

$$r_{i,t} = \sqrt{\tau_t g_{i,t}} \epsilon_{i,t}, \tag{1}$$

where $\tau_t$ and $g_{i,t}$ are, respectively, the long-run and short-run components, while $\epsilon_{i,t}$ is the error term. Due to the high kurtosis of the cryptocurrencies, we assume (as in Amendola et al. 2020) that the error terms follows a Student's $t$ distribution. To take into account the generally high level of skewness of cryptocurrency data, we consider a GJR (Glosten et al. 1993) specification for the short-run component, that is:

$$g_{i,t} = (1 - \alpha - \beta - \frac{\gamma}{2}) + (\alpha + \gamma \mathbb{I}_{(r_{i-1,t}<0)}) \frac{(r_{i-1,t})^2}{\tau_t} + \beta g_{i-1,t}, \tag{2}$$

where $\mathbb{I}_{(\cdot)}$ is an indicator function which equals 1 if the argument is true. It is assumed that $\alpha > 0$ and $\beta \geq 0$, which ensures that $g_{i,t}$ is positive.

The long-run component is as follows:

$$\tau_t = \exp\left\{ m + \theta \sum_{k=1}^{K} \varphi_k(w_1, w_2) X_{t-k} \right\}, \tag{3}$$

where $X$ is the low-frequency variable observed every period $t$ and $\varphi_k(w_1, w_2)$ is the weighting function that is used to weight the $K$ lagged realizations entering Equation (3).

In this work, we use the Beta function as the weighting function in Equation (3). The Beta function is as follows:

$$\varphi_k(w_1, w_2) = \frac{\left(\frac{k}{K+1}\right)^{w_1-1} \left(\frac{1-k}{(K+1)}\right)^{w_2-1}}{\sum_{j=1}^{K} \left(\frac{j}{K+1}\right)^{w_1-1} (1 - j(K+1))^{w_2-1}}. \tag{4}$$

In Equation (4), $w_1$ and $w_2$ must be greater than 1. Fixing $w_1$, the Beta function only requires the estimation of $w_2$ and, more interestingly, allows for a monotonic decreasing scheme for the observations (meaning that more recent observations weight more than more remote ones).

The utilized SB-GARCH-MIDAS model assumes that the exogenous variable $X_t$ may have a structural break. Bearing this in mind, the QLR test is employed on $X_t$ in order to find whether the low-frequency variable has a structural break. If the QLR test detects a structural break, the estimation of the standard GARCH-MIDAS presented above is repeated over two distinct time periods: from the beginning to the date of the break (period one), and from the date following the break to the end of the sample (period two).

Building upon previous research and delving deeper into the critical question of how much of the total variation can be explained by the variation of the long-term component, Engle et al. (2013) provided the variance ratio (VR) in Equation (5):

$$VR = \frac{var(log(\tau_t))}{var(log(\tau_t g_t))}, \tag{5}$$

where $g_t = \sum_{i=1}^{I_t} g_{i,t}$ indicates the summation of the short-term variation concerning the frequency (days within the period: $I_t$) of the exogenous variable in the model (here, monetary policy). The ratio in Equation (5) refers to the variance ratio, which quantifies the relative importance of long-run volatility.

## 3. Results

The dataset contains the daily returns of three cryptocurrencies (Bitcoin, Binance Coin, and XRP). The cryptocurrencies have been chosen based on their popularity, their share in the investment baskets, and their most time-domain coverage. Bitcoin is known as digital gold and designed to be an alternative to traditional currencies. Binance Coin (BNB) launched on the Ethereum platform in 2017 and is known as an asset with fewer trading fees. Furthermore, we employ XRP, which works more like SWIFT. Engineers designed XRP to decrease the costs of transactions with higher speeds. The cryptocurrency data are downloaded from https://coinmarketcap.com, accessed on 30 July 2022. To track the monetary policy in selected countries, we utilized the monthly monetary aggregate (M3), a traditional instrument for economists and decision-makers to direct monetary policy to control inflation, consumption, growth, and liquidity (Chen 2020). Accordingly, the M3 data for the USA and South Africa are collected from the Federal Reserve Economic Data (FRED) archive. Alongside the extensively studied M3 of the USA to stimulate significant comparisons with existing literature, we included the M3 of South Africa. The adoption of phased and structured regulations in South Africa to regulate crypto assets, driven by concerns such as money laundering, terrorism financing, and tax evasion, provides a special restricted-emerging country as a case study (FSB 2022). Furthermore, the South African Reserve Bank (SARB) is also exploring ways to regulate crypto assets after previously adopting a hands-off approach (Singh 2022). The distinct dynamics and triggers of monetary policies and their adjustments encompass a spectrum of economic indicators and considerations driven by a separate set of factors, including inflation, economic growth, and global market conditions, among others. As a result, the model identifies structural breaks based on the most prominent shifts within the specified macroeconomic variable, leading to dissimilar break dates for diverse monetary policies. The identified breaks provide the SB-GARCH-MIDAS models with overlapping sample sizes based on the utilized

macroeconomic variables, underscoring the robustness of our approach in considering multiple macroeconomic variables. Consequently, this design choice avoids the inclusion of other monetary policy instruments in the evaluations, as it prioritizes maintaining a consistent basis for comparison and avoiding potential distortions arising from diverse monetary policy instruments.

       The last observation of the variables is 30 June 2022, while the starting point of the time range for the variables depends on the availability of cryptocurrencies, and it is reported in the last column (Time) of Table 1. To reach the goal and avoid problems such as non-stationarity and asymmetry in the empirical distributions, we use the logarithmic return of the cryptocurrencies and the first difference with one lag of M3's values. The statistical description of the data is reported in Table 1.

**Table 1.** Summary statistics.

|  | N.obs | Min | Mean | STD | Max | Kurtosis | Skewness | Time |
|---|---|---|---|---|---|---|---|---|
| Cryptocurrency |  |  |  |  |  |  |  |  |
| Bitcoin | 3350 | −0.465 | 0.001 | 0.042 | 0.357 | 10.724 | −0.524 | April 2013–July 2022 |
| Binance $_{Coin}$ | 1801 | −0.543 | 0.004 | 0.070 | 0.675 | 15.025 | 0.915 | June 2017–July 2022 |
| XRP | 3252 | −0.616 | 0.001 | 0.071 | 1.027 | 26.791 | 1.577 | August 2013–July 2022 |
| Monetary Aggregate (M3) |  |  |  |  |  |  |  |  |
| USA | 111 | −0.004 | 0.007 | 0.008 | 0.063 | 27.918 | 4.856 | April 2013–July 2022 |
| S. Africa | 111 | −0.013 | 0.006 | 0.007 | 0.033 | 1.185 | 0.384 | April 2013–July 2022 |

N.obs refers to the number of observations for each variable, and STD is the sample standard deviation. The cryptocurrencies are observed daily, while M3 is observed monthly.

       Figures 1 and 2 illustrate the patterns of the log-returns of cryptocurrencies and the first difference divided by one lag of M3 for the USA and South Africa. It is worth noting that the M3 of both the USA and South Africa present structural breaks. We apply the single structural break of the exogenous variable, which involves the long-run component evaluation process. We implement the Wald statistic for the QLR test (Quandt 1960) as a modification of the Chow test (Chow 1960) on a dynamic linear model for the exogenous variable to locate the structural break date. The model is a common practice when the dates for the break are unknown (Éric Racicot and Théoret 2016; Seok et al. 2023; Yang et al. 2023). To reveal the unknown structural breaks, we exclude 15% of observations from each site in the time domain to avoid the short interval for the fitted models in the Chow test. Moreover, we handle the model uncertainty related to the lag selection in the dynamic models by fitting an autoregressive model for the exogenous variable to select the number of lags (Check and Piger 2021). Accordingly, the date of occurrence for the event related to the QRL represents the structural breakpoint, which is applied in the GARCH-MIDAS model by evaluating two GARCH-MIDAS models instead of one according to the detected SB in monetary policy.

       The set of competing models against which we evaluate the proposed SB-GARCH-MIDAS model consists of the standard GARCH model (Bollerslev 1986), the IGARCH model (Engle and Bollerslev 1986), the GJR model (Glosten et al. 1993), and the GARCH-MIDAS model (Engle et al. 2013). All the models have been estimated using, as an error distribution, Student's $t$ distribution with $df$ degrees of freedom. The evaluation of the models uses the Akaike and Bayesian Information Criteria (AIC and BIC, respectively), the Mean Squared Error (MSE) and Quasi-Likelihood (QLIKE) loss functions, as well as the residual diagnostics in terms of the Ljung and Box test (Ljung and Box 1978) on squared standardized residuals. Moreover, we also employ the Model Confidence Set (MCS) procedure (Hansen et al. 2011) to find the set of superior models (SSM), according to the MSE and QLIKE loss functions. The volatility proxy is the squared daily log-returns.

       The final evaluations are reported in Tables 2–4, which are divided into six panels, indicated by A, B, C, D, E, and F. Panel A presents the coefficients for the whole-sample evaluations; Panel B reports the model performance metrics for the whole-sample evaluations;

Panels C and D indicate the model performance metrics for the evaluations regarding the pre- and post-break samples according to the detected structural break in the USA; finally, Panels E and F indicate the model performance metrics for the evaluations regarding the pre- and post-break samples according to the detected structural break in South Africa. Because we have distinguished two sub-samples of pre- and post-break for each country, we have provided the diagnostic tests for the pre- and post-break samples for each country in four panels: C, D, E, and F. When the SB-GARCH-MIDAS considers the period before or after the break, it reduces to the standard GARCH-MIDAS. For these reasons, Panels C, D, E, and F only show the GARCH-MIDAS and not the SB-GARCH-MIDAS. Finally, additional information about the table is provided in the footnote.

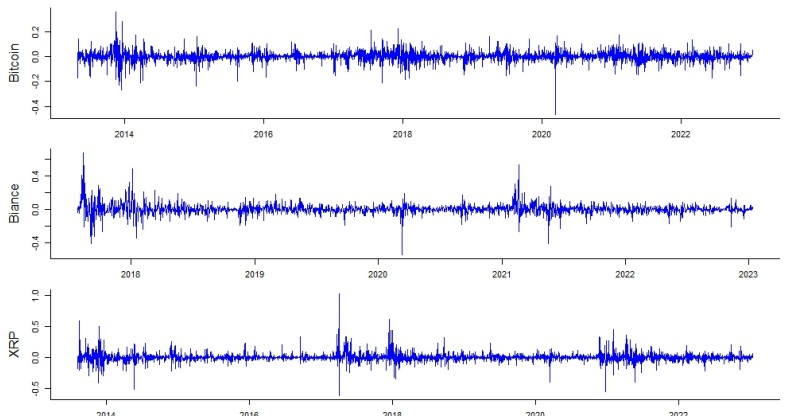

**Figure 1.** Cryptocurrencies (logarithmic returns).

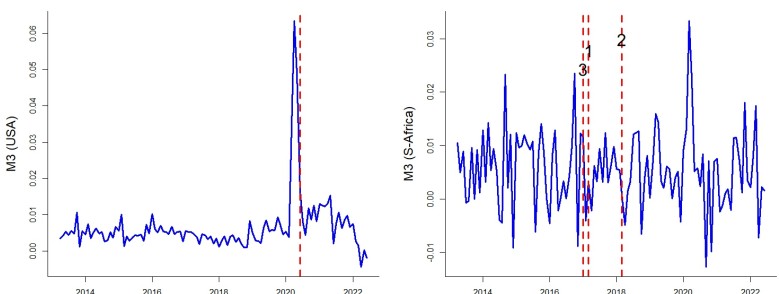

**Figure 2.** Monetary aggregate (M3) for the USA and South Africa (the first difference divided by one lag of M3's values). The vertical lines present the evaluated breakpoints according to the QLR test. The breakpoint of South Africa depends on the selected time domain of the variable concerning the availability of cryptocurrency data (red-line 1: Bitcoin, red-line 2: Binance Coin, red-line 3: XRP).

Table 2 presents the evaluations concerning Bitcoin. We have 3350 daily Bitcoin returns in total, where the post-break observations, respectively, are 761 and 1949 days, according to the break in M3 of the USA and South Africa. Based on the diagnostic tests on panel B for the the whole-sample evaluations, AIC and BIC designate the SB-GARCH-MIDAS models as optimal after incorporating the structural break in the low-frequency values (monthly) of M3 for the USA and South Africa. Using the MSE loss function, the SSM of the MCS test includes GARCH-MIDAS with both the USA's and South Africa's M3 and the proposed SB-GARCH-MIDAS with South Africa's M3. Using the QLIKE loss function, the SSM only consists of the proposed SB-GARCH-MIDAS with the two M3 variables adopted in this work. All of the models have satisfactory residual diagnostics. Finally, all the $\theta$ parameters of the GARCH-MIDAS-based models are largely significant, proving that the M3 is a good long-run volatility driver for Bitcoin.

**Table 2.** Model evaluation for Bitcoin.

| Panel A: The whole-sample evaluations. | | | | | | |
|---|---|---|---|---|---|---|
| | GARCH | GJR | IGARCH | GM M3(USA) | SB-GM M3(USA) | GM M3(SA) | SB-GM M3(SA) |
| *const* | 0.000 | 0.000 | 0.000 *** | | | | |
| | (1.000) | (1.000) | (1.000) | | | | |
| $\alpha_1$ | 0.127 *** | 0.132 *** | 0.127 *** | 0.131 *** | 0.155 *** | 0.133 *** | 0.203 *** |
| | (0.017) | (0.018) | (0.017) | (0.018) | (0.023) | (0.016) | (0.044) |
| $\alpha_2$ | | | | | 0.018 ** | | 0.085 *** |
| | | | | | (0.008) | | (0.015) |
| $\beta_1$ | 0.872 *** | 0.875 *** | 0.873 | 0.878 *** | 0.853 *** | 0.875 *** | 0.819 *** |
| | (0.029) | (0.033) | (1.000) | (0.02) | (0.024) | (0.018) | (0.037) |
| $\beta_2$ | | | | | 0.988 *** | | 0.916 *** |
| | | | | | (0.013) | | (0.02) |
| $\gamma_1$ | | −0.017 | 3.24 *** | −0.022 | −0.02 | −0.019 | −0.048 |
| | | (0.025) | (0.124) | (0.018) | (0.024) | (0.019) | (0.036) |
| $\gamma_2$ | | | | | −0.014 | | −0.01 |
| | | | | | (0.014) | | (0.021) |
| $m_1$ | | | | −4.418 *** | −4.238 *** | −4.49 *** | −4.344 *** |
| | | | | (0.4) | (0.432) | (0.046) | (0.781) |
| $m_2$ | | | | | −5.24 *** | | −4.704 *** |
| | | | | | (0.491) | | (0.305) |
| $\theta_1$ | | | | −25.274 *** | −25.973 *** | 2.646 *** | 42.375 *** |
| | | | | (0.194) | (0.513) | (0.974) | (0.693) |
| $\theta_2$ | | | | | −94.428 *** | | −88.306 *** |
| | | | | | (0.516) | | (14.865) |
| $\omega_{2,1}$ | | | | 3.714 *** | 1.776 *** | 2.034 ** | 6.576 *** |
| | | | | (0.165) | (0.306) | (0.965) | (0.347) |
| $\omega_{2,2}$ | | | | | 1.099 *** | | 1.396 ** |
| | | | | | (0.267) | | (0.704) |
| $df_1$ | 3.25 *** | 3.255 *** | | 3.284 *** | 3.194 *** | 3.268 *** | 2.947 *** |
| | (0.16) | (0.166) | | (0.126) | (0.124) | (0.122) | (0.133) |
| $df_2$ | | | | | 3.493 *** | | 3.408 *** |
| | | | | | (0.516) | | (0.218) |
| **Panel B: Diagnostic tests for the whole sample.** | | | | | | | |
| AIC | −13,374.955 | −13,373.776 | −13,375.815 | −17,207.582 | −17,232.107 | −17,205.766 | −17,244.463 |
| BIC | −13,378.955 | −13,378.776 | −13,379.815 | −17,214.582 | −17,246.107 | −17,212.766 | −17,258.463 |
| MSE | 0.386 | 0.386 | 0.387 | 0.385 | 0.387 | 0.385 | 0.385 |
| QLIKE | −5.529 | −5.524 | −5.528 | −5.525 | −5.532 | −5.524 | −5.531 |
| $LB_{(5)}$ | 0.865 | 0.878 | 0.866 | 0.877 | 0.896 | 0.868 | 0.943 |
| VR | | | | 10.52 | 63.53 | 10.05 | 20.68 |
| *Date of structural break in macro-variable (M3):* | | | | | 31 May 2020 | | 28 February 2017 |
| | | | | | [6.01] | | [42.64] |
| **Panel C: Diagnostic tests for period 1 (pre-break sample) according to the break date of M3 in the USA.** | | | | | | | |
| AIC | −10,425.467 | −10,423.924 | −10,426.135 | −13,387.073 | | | |
| BIC | −10,429.467 | −10,428.924 | −10,430.135 | −13,394.073 | | | |
| MSE | 0.476 | 0.475 | 0.477 | 0.475 | | | |
| QLIKE | −5.509 | −5.505 | −5.509 | −5.506 | | | |
| $LB_{(5)}$ | 0.928 | 0.933 | 0.929 | 0.932 | | | |
| **Panel D: Diagnostic tests for period 2 (post-break sample) according to the break date of M3 in the USA.** | | | | | | | |
| AIC | −2962.947 | −2961.224 | −2963.075 | −3811.421 | | | |
| BIC | −2966.947 | −2966.224 | −2967.075 | −3818.421 | | | |
| MSE | 0.088 | 0.088 | 0.088 | 0.124 | | | |
| QLIKE | −5.608 | −5.607 | −5.607 | −5.412 | | | |
| $LB_{(5)}$ | 0.714 | 0.776 | 0.719 | 0.551 | | | |

**Table 2.** *Cont.*

| Panel E: Diagnostic tests for period 1 (pre-break sample) according to the break date of M3 in South Africa. | | | | |
|---|---|---|---|---|
| AIC | −5955.493 | −5954.629 | −5955.771 | −7562.956 |
| BIC | −5959.493 | −5959.629 | −5959.771 | −7569.956 |
| MSE | 0.389 | 0.392 | 0.39 | 0.395 |
| QLIKE | −5.72 | −5.718 | −5.721 | −5.724 |
| LB$_{(5)}$ | 0.826 | 0.844 | 0.826 | 0.858 |
| Panel F: Diagnostic tests for period 2 (post-break sample) according to the break date of M3 in South Africa. | | | | |
| AIC | −7580.141 | −7578.36 | −7580.559 | −9834.627 |
| BIC | −7584.141 | −7583.36 | −7584.559 | −9841.627 |
| MSE | 0.377 | 0.376 | 0.377 | 0.373 |
| QLIKE | −5.414 | −5.409 | −5.412 | −5.387 |
| LB$_{(5)}$ | 0.792 | 0.796 | 0.793 | 0.787 |

Notes: The table reports the parameters' estimates for Bitcoin. The table is divided into six panels. *Panel A* refers to the whole-sample evaluated coefficients regarding competing models. *Panel B* reports the tests to control the adequacy of whole-sample fitted models. *Panels C* and *E* present the tests to control the adequacy of pre-break sample fitted models concerning the break dates of the USA M3 and South Africa M3. *Panels D* and *F* present the tests to control the adequacy of post-break sample fitted models concerning the break dates of the USA M3 and South Africa M3. Numbers in parentheses are the QMLE standard errors. **, and *** represent the significance at levels 5%, 1%, respectively. The sample period is from 29 April 2013 to 30 June 2022 (3350 daily obs.). LB$_{(5)}$ indicates the *p*-value of the Ljung and Box test (Ljung and Box 1978) on the squared standardized residuals. VR denotes the variance ratio. The light shade of gray indicates the best model according to the AIC and BIC information criteria. Dark shades of gray indicate the models included in the SSM of the MCS procedure, according to the significance level $\alpha = 0.25$. The number in brackets is the maximum F statistic according to the QLR approach.

The GARCH parameters ($\alpha$, $\beta$) are significant and positive, indicating the persistence of short-run and long-run shocks over the Bitcoin market. The long-term component is significant for both the GARCH-MIDAS and SB-GARCH-MIDAS models. However, the sign of the $\theta$ for South Africa changes from positive to negative after the break date. As a result, investors and decision-makers should contemplate the direct impact of the persistence of monetary policy shocks for both the short term and the long term. On the other hand, long-term volatility in Bitcoin returns due to the economic structure in selected countries depends on breakpoints and economic cycles. In addition, the *m* parameter are significant, and we can say that the lags and the elements of the model have been chosen and utilized with sufficient accuracy for our sample. Moreover, variance ratio values, which describe the amount of expected volatility explained by economic variables, are indicated by VR in Panel B of the table. Involving an SB approach in the GARCH-MIDAS model clearly enhances the VR, which demonstrates the importance of considering the structural breaks of economic variables in forecasting volatility in the Bitcoin market.

Finally, Panels C, D, E, and F illustrate the forecasting performance and capability across the competing models. Panels C and D consider the observations before and after 31 May 2020 (structural break in M3 of the USA), and Panels E and F consider the observations before and after 28 February 2017 (structural break in M3 of South Africa). The SSM of the MCS and QLIKE tests include all models for both countries; however, the AIC and BIC detect GARCH-MIDAS as the best models, which indicates the better performance of the models in comparison to competing models.

In the next step, we report the evaluations concerning the Binance Coin returns in Table 3. We have 1801 daily Binance Coin returns in total, where the post-break observations, respectively, are 761 and 1584 days, according to the break in M3 of the USA and South Africa. AIC and BIC detect the SB-GARCH-MIDAS model for South Africa and the GARCH-MIDAS models for the USA, which fit the data. Furthermore, according to the MSE loss function, the SSM of the MCS test includes GARCH-MIDAS with both the USA's M3 and the proposed SB-GARCH-MIDAS with South Africa's M3. The SSM of the QLIKE loss function comprises the proposed SB-GARCH-MIDAS and GARCH-MIDAS, with the two M3 variables adopted in this work. All the models have satisfactory residual diagnostics.

Therefore, the evaluations of the GARCH-MIDAS and SB-GARCH-MIDAS models have priority over other GARCH models in the case of Binance Coin returns.

**Table 3.** Model evaluation for Binance Coin.

| Panel A: The whole-sample evaluations. | | | | | | | |
|---|---|---|---|---|---|---|---|
| | GARCH | GJR | IGARCH | GM M3(USA) | SB-GM M3(USA) | GM M3(SA) | SB-GM M3(SA) |
| *const* | 0.000 *** | 0.000 ** | 0.000 ** | | | | |
| | (1.000) | (1.000) | (1.000) | | | | |
| $\alpha_1$ | 0.184 *** | 0.193 *** | 0.198 *** | 0.176 *** | 0.096 *** | 0.18 *** | 0.318 ** |
| | (0.048) | (0.052) | (0.053) | (0.054) | (0.032) | (0.048) | (0.136) |
| $\alpha_2$ | | | | | 0.173 *** | | 0.15 *** |
| | | | | | (0.049) | | (0.042) |
| $\beta_1$ | 0.802 *** | 0.81 *** | 0.802 | 0.809 *** | 0.916 *** | 0.801 *** | 0.609 *** |
| | (0.044) | (0.049) | (1.000) | (0.065) | (0.032) | (0.059) | (0.191) |
| $\beta_2$ | | | | | 0.778 *** | | 0.766 *** |
| | | | | | (0.062) | | (0.059) |
| $\gamma_1$ | | −0.037 | 3.461 *** | −0.044 | −0.045 * | −0.037 | −0.124 |
| | | (0.041) | (0.242) | (0.033) | (0.027) | (0.035) | (0.148) |
| $\gamma_2$ | | | | | −0.03 | | 0 |
| | | | | | (0.05) | | (0.039) |
| $m_1$ | | | | −5.115 *** | −4.998 *** | −5.073 *** | −4.067 *** |
| | | | | (0.451) | (0.158) | (0.366) | (0.426) |
| $m_2$ | | | | | −5.456 *** | | −5.445 *** |
| | | | | | (0.394) | | (0.248) |
| $\theta_1$ | | | | −22.352 ** | −17.43 *** | −39.769 *** | −2.589 * |
| | | | | (10.534) | (5.146) | (0.153) | (1.481) |
| $\theta_2$ | | | | | −27.996 *** | | −52.299 *** |
| | | | | | (9.185) | | (14.808) |
| $\omega_{2,1}$ | | | | 5.609 *** | 11.84 | 1.981 *** | 1.786 *** |
| | | | | (0.499) | (66.736) | (0.495) | (0.606) |
| $\omega_{2,2}$ | | | | | 5.707 *** | | 1.883 *** |
| | | | | | (0.814) | | (0.395) |
| $df_1$ | 3.593 *** | 3.601 *** | | 3.79 *** | 3.512 *** | 3.793 *** | 4.209 *** |
| | (0.341) | (0.343) | | (0.382) | (0.246) | (0.289) | (1.011) |
| $df_2$ | | | | | 4.295 *** | | 3.775 *** |
| | | | | | (0.559) | | (0.342) |
| **Panel B: Diagnostic tests for the whole sample.** | | | | | | | |
| AIC | −5613.859 | −5612.757 | −5613.511 | −7676.302 | −7667.471 | −7672.958 | −7676.942 |
| BIC | −5617.859 | −5617.757 | −5617.511 | −7683.302 | −7681.471 | −7679.958 | −7690.942 |
| MSE | 3.676 | 3.68 | 3.727 | 3.624 | 3.67 | 3.623 | 3.58 |
| QLIKE | −4.791 | −4.789 | −4.788 | −4.796 | −4.783 | −4.792 | −4.804 |
| $LB_5$ | 0.229 | 0.183 | 0.24 | 0.225 | 0.119 | 0.164 | 0.092 |
| VR | | | | 7.18 | 10.29 | 6.73 | 20.59 |
| *Date of structural break in macro-variable (M3):* | | | | | 31 May 2020 | | 28 February 2018 |
| | | | | | [24.62] | | [8.68] |
| **Panel C: Diagnostic tests for period 1 (pre-break sample) according to the Break date of M3 in the USA.** | | | | | | | |
| AIC | −3089.287 | −3089.156 | −3089.325 | −4277.974 | | | |
| BIC | −3093.287 | −3094.156 | −3093.325 | −4284.974 | | | |
| MSE | 5.096 | 5.16 | 5.101 | 5.078 | | | |
| QLIKE | −4.62 | −4.605 | −4.62 | −4.604 | | | |
| $LB_{(5)}$ | 0.161 | 0.112 | 0.161 | 0.123 | | | |

**Table 3.** *Cont.*

| Panel D: Diagnostic tests for period 2 (post-break sample) according to the break date of M3 in the USA. | | | | | |
|---|---|---|---|---|---|
| AIC | −2644.687 | −2642.8 | −2642.594 | −3551.629 | |
| BIC | −2648.687 | −2647.8 | −2646.594 | −3558.629 | |
| MSE | 1.686 | 1.685 | 1.754 | 1.672 | |
| QLIKE | −5.05 | −5.05 | −5.043 | −5.058 | |
| LB$_{(5)}$ | 0.646 | 0.651 | 0.66 | 0.694 | |
| **Panel E: Diagnostic tests for period 1 (pre-break sample) according to the break date of M3 in South Africa.** | | | | | |
| AIC | −323.15 | −313.887 | −323.148 | | −566.227 |
| BIC | −327.15 | −318.887 | −327.148 | | −573.227 |
| MSE | 20.359 | 19.301 | 20.528 | | 18.585 |
| QLIKE | −3.202 | −3.237 | −3.201 | | −3.217 |
| LB$_{(5)}$ | 0.82 | 0.015 | 0.824 | | 0.831 |
| **Panel F: Diagnostic tests for period 2 (post-break sample) according to the break date of M3 in South Africa.** | | | | | |
| AIC | −5344.646 | −5342.673 | −5340.995 | | −7189.583 |
| BIC | −5348.646 | −5347.673 | −5344.995 | | −7196.583 |
| MSE | 1.531 | 1.532 | 1.595 | | 1.61 |
| QLIKE | −4.992 | −4.993 | −4.98 | | −4.982 |
| LB$_{(5)}$ | 0.052 | 0.051 | 0.056 | | 0.049 |

Notes: The table reports the parameters' estimates for Binance Coin. The table is divided into six panels. *Panel A* refers to the whole-sample evaluated coefficients regarding competing models. *Panel B* reports the tests to control the adequacy of whole-sample fitted models. *Panels C* and *E* present the tests to control the adequacy of pre-break sample fitted models concerning the break dates of the USA M3 and South Africa M3. *Panels D* and *F* present the tests to control the adequacy of post-break sample fitted models concerning the break dates of the USA M3 and South Africa M3. Numbers in parentheses are the QMLE standard errors. *, **, and *** represent the significance at levels 5%, 1%, respectively. The sample period is from 26 July 2017 to 30 June 2022 (1801 daily obs.). VR denotes the variance ratio. LB$_{(5)}$ indicates the *p*-value of the Ljung and Box test (Ljung and Box 1978) on the squared residuals. A light shade of gray indicates the best model according to the AIC and BIC information criteria. Dark shades of gray indicate the models included in the SSM of the MCS procedure, according to the significance level $\alpha = 0.25$. The number in brackets is the maximum F statistic according to the QLR approach.

The short-run and long-run shocks from the monetary policy over the Binance Coin market are persistent and positive. Eventually, all the $\theta$ parameters of the GARCH-MIDAS-based models are significant and negative, which means that the M3 is the good long-run volatility driver for Binance Coin. Compared to the Bitcoin market, the sign of $\theta$ for GARCH-MIDAS and the proposed SB-GARCH-MIDAS models regarding Binance Coin do not depend on breakpoints and economic cycles. Despite the unchanged sign of $\theta$ for Binance Coin, its inception in 2017 considers the economic events until the launch date. In addition, the *m* parameter is significant, which confirms the accuracy specification of the model. According to the VR values, applying the SB approach clearly improves the ability of GARCH-MIDAS models to forecast volatility in the Binance Coin market.

Moreover, the AIC and BIC detect GARCH-MIDAS as the best models according to the evaluations for observations before and after 31 May 2020 (structural break in M3 of the USA) in Panels C and D. Furthermore, the evaluation for observations pre- and post- 28 February 2018 (structural break in M3 of South Africa) in Panels E and F reached the same results about the GARCH-MIDAS model. Therefore, we can indicate the forecasting outperformance capability of the SB-GARCH-MIDAS models in comparison to competing models.

The other important cryptocurrency, which possesses a considerable share of the market, is XRP. We have 3252 daily XRP returns in total, where the post-break observations, respectively, are 761 and 2008 days, according to the break in M3 of the USA and South Africa. In the case of XRP (Table 4), AIC and BIC have proposed the SB-GARCH-MIDAS model for both countries as the best model to fit the data. The SSM of the MCS (in the MSE loss function) only includes the proposed SB-GARCH-MIDAS model with the M3 of South Africa. Using the QLIKE loss function, the SSM consists of GARCH-MIDAS and the proposed SB-GARCH-MIDAS with the two M3 variables. All the models have satisfactory

residual diagnostics. Moreover, the $\theta$ parameters of the GARCH-MIDAS-based models are significant (except for the GARCH-MIDAS with South Africa's M3), proving that the M3 is a good long-run volatility driver for XRP.

**Table 4.** Model evaluation for XRP.

| Panel A: The whole-sample evaluations. | | | | | | | |
|---|---|---|---|---|---|---|---|
| | GARCH | GJR | IGARCH | GM M3(USA) | SB-GM M3(USA) | GM M3(SA) | SB-GM M3(SA) |
| *const* | 0.000 * | 0.000 * | 0.000 *** | | | | |
| | (1.000) | (1.000) | (1.000) | | | | |
| $\alpha_1$ | 0.236 *** | 0.25 *** | 0.236 *** | 0.251 *** | 0.318 *** | 0.247 *** | 0.384 *** |
| | (0.048) | (0.062) | (0.05) | (0.06) | (0.101) | (0.061) | (0.131) |
| $\alpha_2$ | | | | | 0.143 ** | | 0.227 *** |
| | | | | | (0.071) | | (0.079) |
| $\beta_1$ | 0.763 *** | 0.761 *** | 0.764 | 0.76 *** | 0.702 *** | 0.761 *** | 0.65 *** |
| | (0.067) | (0.069) | (1.000) | (0.052) | (0.086) | (0.054) | (0.099) |
| $\beta_2$ | | | | | 0.857 *** | | 0.764 *** |
| | | | | | (0.099) | | (0.083) |
| $\gamma_1$ | | −0.025 | 2.982 *** | −0.031 | −0.049 | −0.026 | −0.079 |
| | | (0.042) | (0.096) | (0.037) | (0.053) | (0.034) | (0.091) |
| $\gamma_2$ | | | | | −0.049 | | −0.025 |
| | | | | | (0.043) | | (0.041) |
| $m_1$ | | | | −3.133 *** | −2.783 *** | −3.232 *** | −3.609 *** |
| | | | | (0.245) | (0.294) | (0.279) | (0.206) |
| $m_2$ | | | | | −4.339 *** | | −3.96 *** |
| | | | | | (0.511) | | (0.423) |
| $\theta_1$ | | | | −19.636 *** | −16.924 *** | −7.302 | 10.942 *** |
| | | | | (1.558) | (1.453) | (5.626) | (4.131) |
| $\theta_2$ | | | | | −68.777 *** | | −89.538 *** |
| | | | | | (1.427) | | (2.3) |
| $\omega_{2,1}$ | | | | 5.818 | 3.058 ** | 2.235 *** | 2.013 *** |
| | | | | (20.956) | (1.346) | (0.142) | (0.263) |
| $\omega_{2,2}$ | | | | | 1.715 | | 1.001 *** |
| | | | | | (1.214) | | (0.197) |
| $df_1$ | 2.986 *** | 2.987 *** | | 2.99 *** | 2.967 *** | 3.001 *** | 3.177 *** |
| | (0.123) | (0.127) | | (0.156) | (0.112) | (0.099) | (0.194) |
| $df_2$ | | | | | 3.141 *** | | 2.925 *** |
| | | | | | (0.233) | | (0.118) |
| **Panel B: Diagnostic tests for the whole sample.** | | | | | | | |
| AIC | −10,763.666 | −10,762.17 | −10,764.031 | −14,484.743 | −14,490.27 | −14,480.541 | −14,490.22 |
| BIC | −10,767.666 | −10,767.17 | −10,768.031 | −14,491.743 | −14,504.27 | −14,487.541 | −14,504.22 |
| MSE | 7.202 | 7.229 | 7.209 | 7.221 | 7.308 | 7.209 | 7.162 |
| QLIKE | −4.773 | −4.775 | −4.773 | −4.782 | −4.781 | −4.778 | −4.769 |
| $LB_{(5)}$ | 0.872 | 0.866 | 0.872 | 0.831 | 0.826 | 0.851 | 0.887 |
| VR | | | | 0.72 | 23.15 | 0.23 | 6.71 |
| *Date of structural break in macro-variable (M3):* | | | | | 31 May 2020 | | 31 December 2016 |
| | | | | | [6.09] | | [37.08] |
| **Panel C: Diagnostic tests for period 1 (pre-break sample) according to the break date of M3 in the USA.** | | | | | | | |
| AIC | −8376.714 | −8375.693 | −8376.979 | −11,226.149 | | | |
| BIC | −8380.714 | −8380.693 | −8380.979 | −11,233.149 | | | |
| MSE | 8.498 | 8.578 | 8.505 | 8.578 | | | |
| QLIKE | −4.806 | −4.81 | −4.806 | −4.814 | | | |
| $LB_{(5)}$ | 0.874 | 0.866 | 0.875 | 0.842 | | | |

**Table 4.** *Cont.*

| Panel D: Diagnostic tests for period 2 (post-break sample) according to the break date of M3 in the USA. | | | | |
|---|---|---|---|---|
| AIC | −2533.104 | −2531.254 | −2533.183 | −3445.654 |
| BIC | −2537.104 | −2536.254 | −2537.183 | −3452.654 |
| MSE | 3.1 | 3.095 | 3.103 | 3.013 |
| QLIKE | −4.711 | −4.715 | −4.711 | −4.743 |
| LB$_{(5)}$ | 0.886 | 0.883 | 0.886 | 0.655 |
| Panel E: Diagnostic tests for period 1 (pre-break sample) according to the break date of M3 in South Africa. | | | | |
| AIC | −4334.846 | −4333.954 | −4335.009 | −5756.526 |
| BIC | −4338.846 | −4338.954 | −4339.009 | −5763.526 |
| MSE | 3.51 | 3.599 | 3.513 | 3.591 |
| QLIKE | −4.824 | −4.835 | −4.824 | −4.829 |
| LB$_{(5)}$ | 0.96 | 0.955 | 0.96 | 0.958 |
| Panel F: Diagnostic tests for period 2 (post-break sample) according to the break date of M3 in South Africa. | | | | |
| AIC | −6590.052 | −6588.334 | −6590.254 | −8922.088 |
| BIC | −6594.052 | −6593.334 | −6594.254 | −8929.088 |
| MSE | 9.378 | 9.397 | 9.387 | 9.203 |
| QLIKE | −4.736 | −4.738 | −4.736 | −4.758 |
| LB$_{(5)}$ | 0.876 | 0.887 | 0.876 | 0.901 |

Notes: The table reports the parameters' estimates for XRP. The table is divided into six panels. *Panel A* refers to the whole-sample evaluated coefficients regarding competing models. *Panel B* reports the tests to control the adequacy of whole-sample fitted models. *Panels C* and *E* present the tests to control the adequacy of pre-break sample fitted models concerning the break dates of the USA M3 and South Africa M3. *Panels D* and *F* present the tests to control the adequacy of post-break sample fitted models concerning the break dates of the USA M3 and South Africa M3. Numbers in parentheses are the QMLE standard errors. *, **, and *** represent the significance at levels 5%, 1%, respectively. The sample period is from 8 August 2013 to 30 June 2022 (3252 daily obs.). VR denotes the variance ratio. LB$_{(5)}$ indicates the *p*-value of the Ljung and Box test (Ljung and Box 1978) on the squared residuals. A light shade of gray indicates the best model according to the AIC and BIC information criteria. Dark shades of gray indicate the models included in the SSM of the MCS procedure, according to the significance level $\alpha = 0.25$. The number in brackets is the maximum F statistic according to the QLR approach.

The GARCH parameters show the persistence of short-run and long-run shocks over the XRP market, except for the $\beta$ value for the IGARCH model. The long-term component is significant for both the GARCH-MIDAS and SB-GARCH-MIDAS models. The *m* values are significant as well. The sign of the $\theta$ in SB-GARCH-MIDAS with South Africa's M3 changes from positive to negative after the structural break date, which is the same as Bitcoin. The comparison between the new case of Binance Coin and the two other senior cryptocurrencies in the study detects how young cryptocurrencies are revised and redesigned to overcome structural breaks. The VR values for the models with XRP returns present obvious improvements in the ability of the SB-GARCH-MIDAS models to forecast the volatility of cryptocurrency markets.

In the end, the AIC and BIC detect GARCH-MIDAS as the best model according to the evaluations for observations after 31 May 2020 (structural break in M3 of the USA) in Panels C and D. Accordingly, the evaluation for observations after 31 December 2016 (structural break in M3 of South Africa) in Panels E and F confirm the better performance of GARCH-MIDAS model. Nevertheless, the SSM of the QLIKE and the MCS includes the GARCH-MIDAS model for both countries according to the sub-samples, which indicates the better forecasting performance ability of the SB-GARCH-MIDAS models in comparison to competing models.

Following the specification of the lag length for the MIDAS model, we provide the Beta weighting scheme (Engle et al. 2013) to detect the effect of the last K (lag) observations. We employ the restricted version by fixing $\omega_1$ to 1. We plot the Beta weighting schemes for the three selected variables in Figure 3. The plot includes nine subplots: the rows define the weighting schemes for high-frequency variables (1st row: Bitcoin, 2nd row: Binance Coin, and 3rd row: XRP), and the columns present the weighting schemes according to the specified country (1st column: the USA and 2nd column: South Africa). Each sub-

figure contains three plots: the dot plots correspond to the GARCH-MIDAS model, the continuous line plots correspond to the SB-GARCH-MIDAS-1 (pre-break sample) model, and the dashed-line plots correspond to the SB-GARCH-MIDAS-2 (post-break sample). Based on the illustrated weighting schemes in Figure 3, all the patterns are very reasonable, and the weights approach zero in a period varying from nine months (for Binance Coin and USA M3 as an additional low-frequency variable, for instance) to 24 months (for XRP and USA M3 as an additional low-frequency variable, for instance).

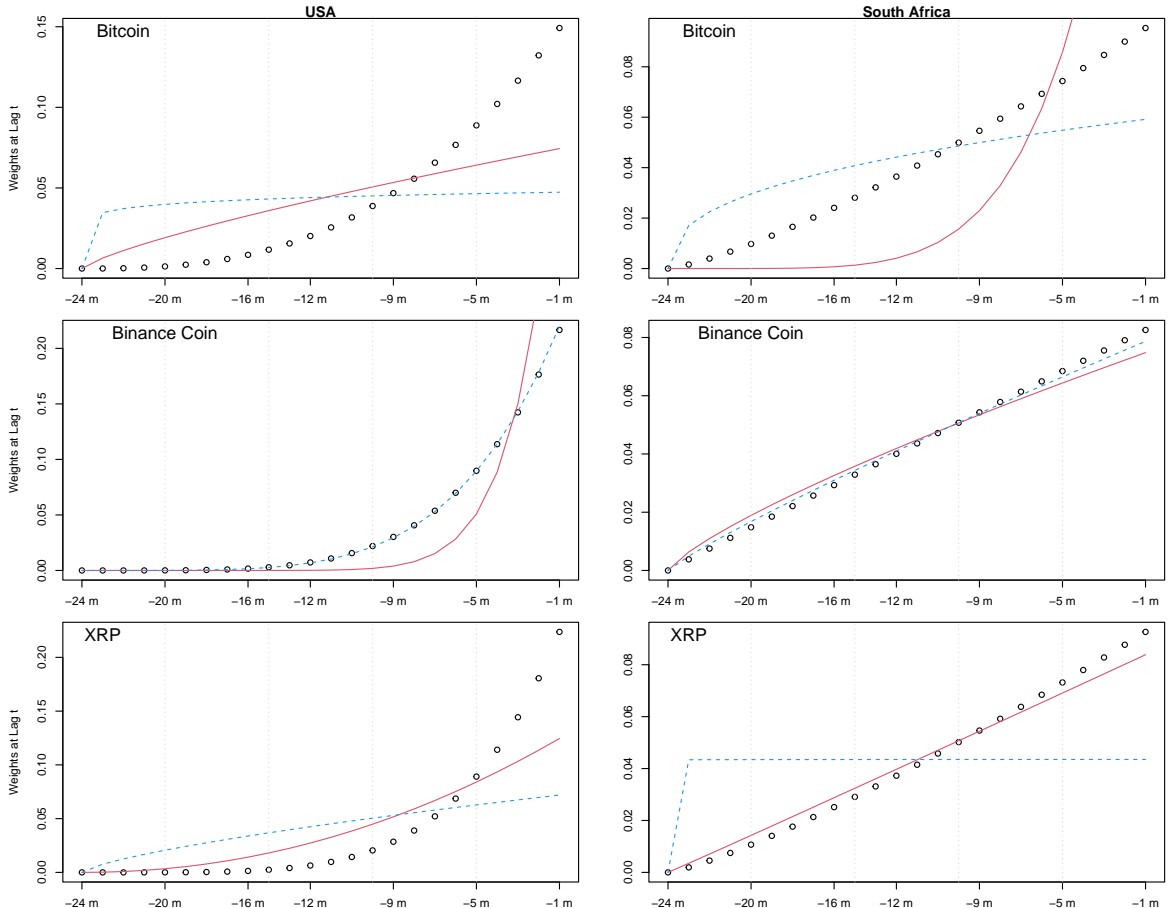

**Figure 3.** Plots for Beta weighting schemes for Bitcoin returns (**first row**), Binance Coin returns (**second row**), and XRP returns (**third row**) for the whole-sample period according to the monetary policy of the USA (**first column**) and the monetary policy of South Africa (**second column**). The dot plots correspond to the weighting schemes for the GARCH-MIDAS model, the continuous line plots correspond to the weighting schemes for the SB-GARCH-MIDAS-1 (pre-break sample) model, and the weighting schemes for the dashed-line plots correspond to the SB-GARCH-MIDAS-2 (post-break sample) model.

## 4. Discussion

The rapid advancement of technology and the subsequent creation of new opportunities in line with the application of monetary and financial instruments directed cryptocurrencies to take on the traditional features of money. Cryptocurrencies represent a burgeoning market that continues to attract a growing number of individuals and organizations, with rapidly changing valuations and deepening connections to other economic variables. This dynamism introduces potential risks to the broader economy that necessitate careful monitoring and analysis. Therefore, the study examines the evolving landscape of cryptocurrencies in the context of monetary policy in addition to the volatility forecast performance of the evaluated models. We proposed the SB-GARCH-MIDAS approach to detect the impact of monetary policy on cryptocurrencies and their long-term volatility.

Our study covered a comprehensive dataset comprising monetary policy indicators for the USA and South Africa, along with daily returns data for three prominent cryptocurrencies: Bitcoin, Binance Coin, and XRP.

The empirical findings of our research have yielded valuable insights. We use the AIC and BIC, in addition to the MSE and QLIKE loss functions and the residual diagnostics in terms of the Ljung and Box test (Ljung and Box 1978) on squared standardized residuals to evaluate the proposed SB-GARCH-MIDAS and GARCH-MIDAS models. Moreover, we take advantage of the MCS procedure (Hansen et al. 2011) to find the SSM, according to the MSE and QLIKE loss functions. The volatility proxy is the squared daily log returns. The SSM of the MSE and QLIKE loss functions presents evidence that the proposed SB-GARCH-MIDAS outperforms the other GARCH model specifications.

The study shows that importing monetary policy into the long-term volatility equations raises the credibility of the GARCH models in volatility prediction. In addition, the application of the Structural Break model detects that the sign of long-term volatility varies around the structural breakpoint for senior and established cryptocurrencies (Bitcoin and XRP), while it is the same pattern for emerging cryptocurrencies (Binance Coin).

Cryptocurrencies are a growing market that involves an increasing number of people and organizations with rapid evaluations and increasing interconnections to other economic variables, which could be a risk for the whole economy without an applicable abstraction. Our study serves as a valuable resource for decision-makers, offering insights into the interplay between cryptocurrencies and macroeconomic determinants. Moreover, our inclusion of a diverse range of cryptocurrencies, spanning both established and emerging types, illustrates how the nature of the cryptocurrency can influence correlations and volatility, enriching our understanding of this complex ecosystem.

Cryptocurrencies are not homogeneous; they differ according to technology and type, which may affect the variety of research findings. Additionally, the temporal dimension of our models captures a broad spectrum of information, although not all fluctuations over the series are necessarily accounted for due to temporal variations. The other important point is the time domain of the models. In the current study, we tried to cover the most possible information. There are some fluctuations over the series; however, these events have not affected all models due to time differences.

Our study holds significant implications for investors and decision-makers in the cryptocurrency space, which is assuming a more prominent role in the global financial landscape. Understanding their behavior and vulnerabilities becomes crucial for strategic decision-making. By examining the impact of monetary policy on long-term volatility in cryptocurrencies inside the proposed SB-GARCH-MIDAS model, the study provides and offers investors a more robust framework for predicting volatility. In addition, the findings highlight the diverse nature of cryptocurrencies and their potential risks to the broader economy, empowering decision-makers with the knowledge needed to navigate this rapidly evolving market and formulate informed investment and risk management strategies.

The study provides further exploration and research avenues. The dynamic essence of cryptocurrencies and their interactions with monetary policy and other economic elements offer immense opportunities for future investigations. Researchers can delve deeper into the nuances of cryptocurrency behavior across distinct types and under varying economic conditions, eventually advancing our understanding of this rapidly evolving digital financial landscape. We make a contribution to a deeper understanding of the complex processes at play in this growing and transformational domain by fusing rigorous analytical methodologies with varied datasets. It is worth mentioning that, in line with the enhancement of statistical methods, the study sheds light on the impact of country-specific monetary decisions, such as those in the USA and South Africa, on cryptocurrencies. However, a more extensive country-versus-country comparison of models focusing on controlling international effects could provide deeper insights into the distinct regulatory landscapes and economic factors shaping the dynamics of cryptocurrencies. This avenue presents an exciting opportunity for future investigations, allowing for a more compre-

hensive understanding of the intricate interplay between individual countries and global digital assets.

**Author Contributions:** All authors contributed substantially to this work. Conceptualization, V.C. and S.D.J.; methodology, A.A., V.C. and S.D.J.; software, V.C. and S.D.J.; validation, M.S.A., A.A., V.C. and S.D.J.; formal analysis, S.D.J.; investigation, S.D.J.; resources, S.D.J.; data curation, S.D.J.; writing—original draft preparation, S.D.J.; writing—review and editing, M.S.A., A.A., V.C. and Sh.D.J.; visualization, V.C. and S.D.J.; supervision, M.S.A. and A.A.; project administration, M.S.A., A.A. and V.C. All authors have read and agreed to the published version of this manuscript.

**Funding:** The authors gratefully acknowledge funding from the Italian Ministry of Education University and Research MIUR through PRIN project *Econometric Methods and Models for the Analysis of Big Data: Improving Forecasting Ability by Understanding and Modelling Complexity* [grant number 201742PW2W_002].

**Institutional Review Board Statement:** Not applicable.

**Informed Consent Statement:** Not applicable.

**Data Availability Statement:** The data in this paper were retrieved from Coinmarketcap https://coinmarketcap.com/, (accessed on 30 June 2022) and the Federal Reserve Economic Data (FRED) archive https://fred.stlouisfed.org/, (accessed on 30 June 2022).

**Conflicts of Interest:** The authors declare no conflict of interest.

## Abbreviations

The following abbreviations are used in this manuscript:

| | |
|---|---|
| AIC | Akaike Information Criteria |
| AR | AutoRegressive |
| BIC | Bayesian Information Criteria |
| CBDCU | Central Bank Digital Currency Uncertainty Index |
| ECB | European Central Bank |
| EPU | Economic Policy Uncertainty |
| FRED | Federal Reserve Economic Data |
| GARCH | Generalized AutoRegressive Conditional Heteroskedasticity |
| MCS | Model Confidence Set |
| MDPI | Multidisciplinary Digital Publishing Institute |
| MIDAS | Mixed Data Sampling |
| MSE | Mean Squared Error |
| OMO | Open Market Operations |
| PoB | Proof-of-Burn |
| PoW | Proof-of-Work |
| QLIKE | Quasi-Likelihood |
| QLR | Quandt Likelihood Ratio |
| SB | Structural Break |
| SSM | Set of Superior Models |
| VAR | Vector AutoRegressive |
| VR | Variance Ratio |

## Note

[1] Alexakis et al. (2024) considered 93 events related to the limitation of fiat currency circulation.

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
