# Peer review of "Is Monetary Policy a Driver of Cryptocurrencies? Evidence from a Structural Break GARCH-MIDAS Approach"

_econometrics, doi:10.3390/econometrics12010002_

Round 1
Reviewer 1 Report
Comments and Suggestions for Authors
Several comments have been provided in the text to improve the quality of the research work. All the comments have been marked in the PDF file attached herewith. Please have a look.

The quality of English of the manuscript, particularly the INTRODUCTION SECTION should be improved.
Reviewer 2 Report
Comments and Suggestions for Authors
Please find my referee report in the attachment.

No major issues detected, although I would suggest the authors to do a proofread of all the sentences as some of them read a bit awkward.
Reviewer 3 Report
Comments and Suggestions for Authors
p. 4 lines 181-184 and p.6 lines 233-239. The potential impact of the implementation of phased and structured regulation for handling crypto assets in South Africa on the prices and volatility of cryptocurrencies is understandable. Meanwhile, I would propose to elaborate more on how these regulation changes are incorporated in M3 data because only M3 data is used in the model.
p. 4 line 144, p.8 line 274 Taking into account references to (Engle et al. 2013) It is proposed to address in the paper the same question as in (Engle et al. 2013) “How much of expected volatility can be explained by economic variables?”
p. 4 lines 180-190 p. 6 lines 229-239. The authors do discuss the selection countries for modelling. Meanwhile, taking into account that the paper shows that the country-specific monetary decisions taken in the country that have a limited global impact in many aspects have a material impact on the global digital asset (Bitcoin). It is proposed to elaborate more on country vs. country comparison of models and controlling of international effects, for example, see the approach in Corbet et al. (2017). In particular, why does South Africa M3 data work better than USA M3 data (p.8 lines 284-285, p.9 lines 302-303)?
Round 2
Reviewer 2 Report
Comments and Suggestions for Authors
I would like to thank the authors to making the effort to conduct a comprehensive revision to the manuscript. My comments from the previous review round have been adequately addressed.
Reviewer 3 Report
Comments and Suggestions for Authors
Dear Authors,
I have highly appreciated the implemented changes and provided a well-structured reply to the comments.
Good luck!